

# HD-6mAPred: a hybrid deep learning approach for accurate prediction of N6-methyladenine sites in plant species

Huimin Li*, Wei Gao*, Yi Tang and Xiaotian Guo

School of Mathematics and Computer Science, Yunnan Minzu University, Kunming, Yunnan, China
* These authors contributed equally to this work.

## ABSTRACT

**Background:** N6-methyladenine (6mA) is an important DNA methylation modification that serves a crucial function in various biological activities. Accurate prediction of 6mA sites is essential for elucidating its biological function and underlying mechanism. Although existing methods have achieved great success, there remains a pressing need for improved prediction accuracy and generalization cap ability across diverse species. This study aimed to develop a robust method to address these challenges.

**Methods:** We proposed HD-6mAPred, a hybrid deep learning model that combines bidirectional gated recurrent unit (BiGRU), convolutional neural network (CNN) and attention mechanism, along with various DNA sequence coding schemes. Firstly, DNA sequences were encoded using four different ways: one-hot encoding, electron-ion interaction pseudo-potential (EIIP), enhanced nucleic acid composition (ENAC) and nucleotide chemical properties (NCP). Secondly, a hold-out search strategy was employed to identify the optimal features or feature combinations for both BiGRU and CNN. Finally, the attention mechanism was introduced to weigh the importance of features derived from the BiGRU and CNN.

**Results:** A series of experiments on the *Rosaceae*, rice and *Arabidopsis* datasets were conducted to demonstrate the superiority of HD-6mAPred. In *Rosaceae*, the HD-6mAPred model achieved excellent performance: accuracy (ACC) of 0.996, Matthew correlation coefficient (MCC) of 0.993, sensitivity (SN) and specificity (SP) of 0.995 and 0.998, respectively. In rice, the evaluation metrics are 0.952 (ACC), 0.905 (MCC), 0.955 (SN), and 0.949 (SP). In *Arabidopsis*, the corresponding metrics are 0.937 (ACC), 0.875 (MCC), 0.927 (SN), and 0.948 (SP). Compared to existing methods, these results demonstrate that HD-6mAPred achieves state-of-the-art performance in predicting 6mA sites across three plant species. Furthermore, HD-6mAPred not only improves the accuracy of 6mA site prediction, but also shows excellent generalization capability across species. The source code utilized in this study is publicly accessible at https://doi.org/10.5281/zenodo.15355131.

Corresponding author
Huimin Li, lihuimin_1980@126.com

## INTRODUCTION

DNA methylation is involved in the processes of species formation and evolution (*Vanyushin et al., 1968*). DNA N6-methyladenine (6mA) refers to an epigenetic modification characterized by the incorporation of a methyl group to the N6 atom of adenine in the DNA molecule. It is a significant methylation modification, and was first reported clearly in eukaryotes by *Gorovsky, Hattman & Pleger (1973)*. This modification is essential for various biological functions, including gene regulation (*Xie et al., 2020*), DNA replication (*Costa & Diffley, 2022*), repair (*Awwad et al., 2023*) and inheritance (*Lee et al., 2023*). Moreover, 6mA has been related to numerous diseases, such as cancer, neurological disorders, metabolic disorders, immune-related diseases, and inflammation (*Lin et al., 2022*). Therefore, accurately identifying 6mA sites is crucial for elucidating the role and mechanism of 6mA epigenetic modification.

Currently, several wet experiments have been designed to identify 6mA sites in DNA, such as liquid chromatography and tandem mass spectrometry (LC-MS/MS) (*Ye et al., 2023*), SMRT sequencing (SMRT-seq) (*Adhikari, Erill & Curtis, 2021*), restriction enzyme-based sequencing (6mA-REseq) (*Li et al., 2022*) and 6mA-IPseq (*Zhang et al., 2018*). While LC-MS/MS (*Ye et al., 2023*) allows for precise mass analysis of nucleosides and can identify and quantify 6mA, it is limited to detecting modified adenosine present in specific target motifs. Despite its high sensitivity for abundant 6mA, SMRT-seq suffers from relatively lower specificity and cannot reliably distinguish between 6mA and m1A (*Luo et al., 2015*). Additionally, 6mA-REseq (*Li et al., 2022*) is constrained by specific restriction sites and exhibits a high false positive rate. Similarly, 6mA-IPseq (*Li et al., 2022*) faces challenges with low specificity and relatively low sensitivity, also struggling to differentiate between m1A, 6mA and reliably enriching the unmethylated DNA fragments. Furthermore, these methods are commonly time-consuming, labor-intensive and expensive, which poses challenges for high-throughput detection of 6mA sites.

To overcome the limitations of biological experiments, several machine learning (ML) algorithms have been introduced to detect 6mA modifications in plants, exemplified by i6mA-pred (*Chen et al., 2019*), i6mA-Fuse (*Hasan et al., 2020*), Meta-i6mA (*Hasan et al., 2021*), i6mA-stack (*Khanal et al., 2021*), i6mA-vote (*Teng et al., 2022*) and 6mA-StackingCV (*Huang, Huang & Luo, 2023*). The success of these approaches is largely determined by two critical factors: feature extraction and the chosen ML algorithms. For example, i6mA-Fuse (*Hasan et al., 2020*) is the first computational model designed to predict 6mA sites in the *Rosaceae* genomes, employing five different encoding schemes to construct five random forest (RF) models. Meta-i6mA (*Hasan et al., 2021*) used a two-stage feature selection process to identify optimal characteristics and leveraged six different ML methods including RF, support vector machine (SVM), extreme random tree (ERT), logistic regression (LR), naive Bayes (NB) and AdaBoost to establish baseline models in the *Rosaceae* genome. In i6mA-vote (*Teng et al., 2022*), one-hot encoding, a widely used method for encoding DNA or RNA sequences (*Alipanahi et al., 2015*; *Kelley, Snoek & Rinn, 2016*; *Zhou & Troyanskaya, 2015*) is used for dinucleotides, and an ensemble learning framework is constructed through the incorporation of the five ML

methods to predict 6mA in *Rosaceae*, rice and *Arabidopsis*. 6mA-StackingCV (*Huang, Huang & Luo, 2023*) employed a hold-out validation strategy to evaluate the effectiveness of feature combination and ML-based ensembles. i6mA-stack (*Khanal et al., 2021*) implemented a recursive feature elimination approach for extracting the optimal feature combination and selected SVM as a meta-classifier alongside a combination of SVM, LR, RF and Gaussian naive Bayes as base-classifiers to predict m6A sites in *F. vesca* and *R. chine*nsis. 6mA-Finder (*Xu et al., 2020*) also adopted the similar feature selection methods with i6mA-stack and demonstrated that RF outperformed other classifiers in 6mA predictions specific to mice and rice. While these methods have yielded promising results, they primarily rely on traditional ML algorithms that need handcrafted features, making them susceptible to noise.

Deep learning methodologies possess the ability to autonomously identify more effective characteristics from original datasets. In the realm of relevant research, a number of deep-learning methods have been put forward with the aim of accurately predicting various methylation sites, among which 6mA sites is included (*Liu et al., 2019*; *Rehman et al., 2022*; *Tang et al., 2022*; *Tsukiyama et al., 2022*; *Tsukiyama, Hasan & Kurata, 2022*; *Nguyen-Vo et al., 2023*; *Nguyen-Vo, Rahardja & Nguyen, 2024*; *Yang et al., 2024*). i6mA-Caps (*Rehman et al., 2022*) used a single encoding scheme for DNA sequence numerical representation, with convolution layers extracting low-level features from the numerical data and a capsule network using these to extract intermediate- and high-level features for 6mA sites classification on *Rosaceae*, rice and *Arabidopsis thaliana*. CNN6mA (*Tsukiyama, Hasan & Kurata, 2022*) is an interpretable neural network model based on position-specific CNN and cross-interactive network for 6mA site prediction in 11 species. BERT6mA (*Tsukiyama, Hasan & Kurata, 2022*) ultilized a hybrid models including the BERT with Bi-LSTM and the 1D-CNN with Bi-LSTM to predict 6mA sites also in 11 species. i6mA-CNN (*Nguyen-Vo, Rahardja & Nguyen, 2024*) employed convolutional neural networks with a fusion of multiple receptive field to identify 6mA sites in mouse genomes. iDNA6mA-Rice (*Lv et al., 2019*) is an advanced computational model utilizing deep learning methods, which encodes and extracts essential genomic characteristics using an embedding layer, as well as multiple fully connected layers. Notably, while MM-6mAPred (*Pian et al., 2020*) introduced a Markov model-based classification method, utilizing the transfer probabilities between adjacent nucleotides to identify 6mA sites, its feature processing ideas could be effectively combined with deep learning approaches. Overall, these methods have demonstrated effective results, providing valuable insights for subsequent research and establishing a foundation for this study. However, they also exhibit certain limitations, such as insufficient generalization across different species, a need for improved accuracy and opportunities to enhance feature extraction methodologies.

In view of this, we introduce an ensemble model, termed HD-6mAPred, that integrates bidirectional gated recurrent unit (BiGRU), convolutional neural network (CNN) and attention mechanism to enhance the prediction performance of DNA 6mA sites by employing multiple encoding strategies. Firstly, we compared the accuracy (ACC) values of single encodings and their corresponding combinations using the BiGRU and CNN

models based on one-hot encoding (*Alipanahi et al., 2015*; *Kelley, Snoek & Rinn, 2016*; *Zhou & Troyanskaya, 2015*), electron-ion interaction pseudo-potential (EIIP) (*Alakuş, 2023*), enhanced nucleic acid composition (ENAC) (*Huang et al., 2018*) and nucleotide chemical properties (NCP) (*Chen et al., 2015*). Secondly, based on the obtained ACC values, CNN employed a joint encoding strategy of NCP (*Chen et al., 2015*) and ENAC (*Huang et al., 2018*), while BiGRU utilizes the combination of ENAC, one-hot and NCP encodings. Finally, the outputs of the BiGRU and CNN were fused using an attention mechanism to test the importance of features. In terms of encoding ways, HD-6mAPred considers the physical and chemical properties of nucleotides, as well as their local composition and sequence information. Structurally, HD-6mAPred not only addresses long-range dependencies within the sequence but also pays attention to the extraction of local features, while accounting for the importance of different features. A series of experimental results demonstrate the excellent performance of HD-6mAPred, in predicting 6mA sites.

## MATERIALS AND METHODS

### Datasets

High-quality datasets are essential for model construction and evaluation. For the convenience of the comparison, we utilized the same datasets which are previously employed in 6mA prediction (6mA-StackingCV (*Huang, Huang & Luo, 2023*) and i6mA-vote (*Teng et al., 2022*). The datasets include: the *Rosaceae* training dataset, the *Rosaceae* testing dataset, the Rice dataset, and the *Arabidopsis* dataset. In which, the *Rosaceae* training dataset is employed to train the model and debug parameters, and the other three datasets, also named independent testing sets, are utilized to test the performance of the model. The Rice dataset is built by *Lv et al. (2019)*, whereas the *Rosaceae* and *Arabidopsis* datasets are curated by *Hasan et al. (2021)*. In summary, the *Rosaceae* training dataset contains 29,237 positive samples (6mA site-containing sequences) and 29,433 negative samples (non 6mA site-containing sequences), and the *Rosaceae* testing dataset consists of 7,298 positive samples and 7,300 negative samples. The numbers of positive samples and negative samples are 153,635 and 153,629 in the Rice dataset, respectively; and they are 31,414 and 31,843 in the *Arabidopsis* dataset. Each sequence is 41 bp long, with nucleobase A located at position 21. The information of these datasets has been given in 6mA-StackingCV (*Huang, Huang & Luo, 2023*), and the specific data can be downloaded from https://github.com/Xiaohong-source/6mA-stackingCV provided in the article.

### Feature extraction

Feature selection is vital to the effectiveness of a predictive model. We explored four different feature encoding ways including one-hot encoding (*Alipanahi et al., 2015*; *Kelley, Snoek & Rinn, 2016*; *Zhou & Troyanskaya, 2015*), EIIP (*Alakuş, 2023*), NCP (*Chen et al., 2015*) and ENAC (*Huang et al., 2018*), which are widely used to represent DNA sequences for predicting 6mA sites.

In one-hot encoding, each nucleotide is encoded as a 4-dimensional binary vector, *i.e.*, nucleotides A, C, G, and T, respectively denoted as A = [1, 0, 0, 0], C = [0, 1, 0, 0],

G = [0, 0, 1, 0], and T = [0, 0, 0, 1] (*Alipanahi et al., 2015*; *Kelley, Snoek & Rinn, 2016*; *Zhou & Troyanskaya, 2015*). EIIP expresses nucleotides according to their electron–ion energy distribution in a given DNA sequence, and the four nucleotides are respectively represented as A = 0.1260, C = 0.1340, G = 0.0806 and T = 0.1335 (*Alakuş, 2023*). NCP encodes nucleotides based on their chemical properties, and the four nucleotides are expressed as A = [1, 1, 1], C = [0, 1, 0], G = [1, 0, 0] and T = [0, 0, 1], respectively (*Chen et al., 2015*). ENAC is the frequency of occurrence of a nucleotide in a sequence starting from the 5′ end to the 3′ end, determined through

$$V = \left[ \frac{N_{A,\,win1}}{S}, \frac{N_{C,\,win1}}{S}, \frac{N_{G,\,win1}}{S}, \frac{N_{T,\,win1}}{S}, \ldots, \frac{N_{A,\,winL-S+1}}{S}, \frac{N_{C,\,winL-S+1}}{S}, \right.$$
$$\left. \frac{N_{G,\,winL-S+1}}{S}, \frac{N_{T,\,winL-S+1}}{S} \right]$$

where S denotes the width of the sliding window (*Huang et al., 2018*) and it equals five in this study, $N_{M,\,winj}$ represents the occurrence number of nucleotide M in the window j, $M \in \{A, C, G, T\}$, j = 1, 2, …., L-S+1.

To obtain the optimal combination of features, we implemented a hold-out search strategy similar to that used in 6mA-StackingCV (*Huang, Huang & Luo, 2023*). This strategy can test the performance of various feature combinations in distinguishing 6mA sites from non-6mA sites. Unlike 6mA-StackingCV (*Huang, Huang & Luo, 2023*), which aims to identify better features within a conventional ML framework, we employed BiGRU and CNN as our learning algorithms to achieve improved performance with the deep learning paradigm. To illustrate the generalization of selected features, we only carried out a series of experiments on the *Rosaceae* training dataset. The *Rosaceae* training dataset was split into the hold out-training dataset and hold out-testing dataset as a ratio of 8:2. First, we computed the ACC of the BiGRU using a single feature encoding and selected the feature with the largest ACC value as the first baseline feature, named as the first-order encoding. Second, we combined the first-order encoding with the remaining three single encoding ways. The increasing ACC values mean that the added feature encoding contributed positively to the performance. If there exist ACC values that are more than that of the first-order encoding, then the combination having the highest ACC score is chosen as the updated encoding, denoted as the second-order encoding. The process continued, adding remaining single encoding ways to the current combination, until no further ACC improvements were observed. The feature combination with the highest ACC value is used to construct the 6mA sites predictor. The strategy for selecting the optimal encoding ways under the CNN is the same as above.

## Deep learning approach

As illustrated in Fig. 1, in constructing the model, we mainly employed three deep learning modules: BiGRU, CNN and attention mechanism. As an improved form of recurrent neural network (RNN), GRU can address the vanishing gradient problem often encountered in the processing of long sequences and has demonstrated excellent performance in addressing sequential challenges. Conversely, CNN excels at achieving

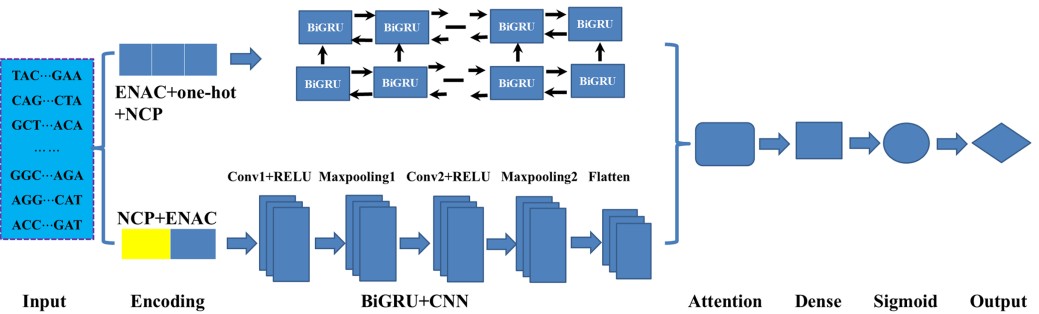

**Figure 1 An overview of the proposed HD-6mAPred.** Conv1 and Conv2 represent the first and the second convolutional layers, respectively; Maxpooling1 and maxpooling2 denote the first and the second max pooling layers, respectively. The term BiGRU+CNN represents the fusion module integrating BiGRU and CNN.

parallel computation and capturing local dependencies in data (*Wang et al., 2024*). To effectively integrate the outputs from both networks, we employed an attention mechanism that fuses features extracted by the BiGRU and CNN. Firstly, the sequences were fed to the BiGRU and CNN modules according to their corresponding optimal encoding ways. Subsequently, the attention mechanism assigns weights to the features extracted by both the BiGRU and CNN, highlighting important features and filtering out less relevant information. Finally, the features refined by the attention mechanism were processed through a fully connected layer (Dense) and a sigmoid activation function, resulting in a score between 0 and 1 in identifying the DNA 6mA sites. The details of the methods involved in the model were listed follows.

BiGRU: In DNA sequence data, the BiGRU module is designed to capture both forward and backward dependencies (*Nguyen-Vo et al., 2023*; *Jia et al., 2024*). It is composed of forward and backward GRUs, which read the sequences from two opposite directions. Special gating mechanisms regulate the retention and forgetting of information from previous states. These mechanisms generate candidate hidden states, which are then used to calculate the current hidden state. Finally, the outputs of the forward and backward GRUs are concatenated along the feature dimension to obtain a comprehensive feature representation.

CNN: CNN utilizes convolution and pooling operations to extract essential local features. Through the introduction of non-linearity using the ReLU activation function, the network can efficiently reduce data dimensions *via* pooling. In this study, the computation can be expressed as follows Eqs. (1)–(3):

$$Y[i] = \sum_j W[j].X[i+j] \tag{1}$$

$$RELU(x) = \max(0, x) \tag{2}$$

$$Y_{pool}[i] = \max(X[i:i+k]) \tag{3}$$

where X is the input sequence, W denotes the convolution kernel, Y[i] represents the convolution output, W[j] is the convolution weight, X[i + j] is the local region of the input

sequence. k is the size of the pooling window, and $Y_{pool}[i]$ is the maximum pooling. While the max function selects the maximum value from the defined window, effectively reducing feature dimensions.

Attention mechanism: The essence of the attention mechanism is to calculate the values representing the significance of different components. It comprises three components: Query (Q), Key (K) and Value (V). The similarity of Q and K is usually quantified through dot products:

$$score(Q, K) = Q \cdot K^T \tag{4}$$

$$Attention = softmax\left(\frac{Q \cdot K^T}{\sqrt{d_k}}\right)V \tag{5}$$

$$Q = W_Q \cdot h_{GRU} \tag{6}$$

$$K = W_K \cdot h_{CNN} \tag{7}$$

$$V = W_V \cdot h_{CNN} \tag{8}$$

where $d_k$ denotes the size of the K vector, softmax is used to normalize the weights. $W_Q$, $W_K$ and $W_V$ are the learned weight matrices. In our model, the Query originates from the output of BiGRU, while the Key and Value are derived from the output of CNN.

Output layer: The ultimate result is generated by a dense layer that maps the outcomes of the attention mechanism to the final predictions as Eq. (9).

$$y_{pred} = \sigma(W_o \cdot attention + b_o) \tag{9}$$

where $W_o$ represents the weight matrix in the output layer, $b_o$ means the bias term, and $\sigma$ denotes the sigmoid function.

Loss function: The model employs binary cross entropy (*Ma, Liu & Qian, 2004*) as its loss function, appropriate for binary classification tasks, and the formula is as follows:

$$loss = -\frac{1}{N}\sum_{i=1}^{N}\left[y_i \log\left(y_{pred,i}\right) + (1 - y_i)\log\left(1 - y_{pred,i}\right)\right] \tag{10}$$

where $y_i$ is the true label and $y_{pred,i}$ is the predicted probability.

## Detail procedure and hyperparameters setting

In this section, we give the detail procedure and hyperaramters about HD-6mAPred.

For the BiGRU module, the number of BiGRU layers is set to 2, with 64 units per layer. The activation function is tanh, and the gated activation function is sigmoid. The dropout rate is 0.5.

For the CNN module, the number of convolutional layers is 2, with 128 filters per layer. The kernel size is 3, and the activation function is the rectified linear unit (ReLU). After each convolutional layer, the max pooling layer has a pooling size of 2. After the two convolutional operations, a flatten layer converts the multi-dimensional output into a one-dimensional vector, with an additional dropout rate of 0.5.

The outputs of the two subnetworks are connected through the concatenate layer, and the attention mechanism is used for feature weighting. The dimension of query, key, and value in the attention mechanism is 16. The output of attention is processed through layer normalization, and the final output is obtained from a fully connected layer (Dense), whose activation function is sigmoid.

For model optimization, the optimizer is Adam, and the initial learning rate is 0.0001. We utilized a 5-fold cross-validation method for training and testing the model. During each fold's training phase, we set the maximum number of epochs to 100 and the batch size to 64. An early stopping strategy was implemented to monitor the validation set loss. Training ceases if there is no improvement after five consecutive epochs, and the model reverts to the optimal weights obtained.

### Evaluation metrics

To test the performance of HD-6mAPred, we implemented a 5-fold cross-validation. Specifically, all samples were partitioned into five groups randomly, the four of which were used to train the model, while the remaining one was designated for testing the model. This procedure is repeated five times to ensure that each sample serves as a testing set at least once. The final testing results were obtained by averaging the metrics from the five iterations. Four standard evaluation metrics consisting of accuracy (ACC), Matthews correlation coefficient (MCC), sensitivity (SN) and specificity (SP) are employed for evaluating the performance of HD-6mAPred. These metrics can be formulated by Eqs. (11)–(14).

$$ACC = \frac{TP + TN}{TN + FP + TP + FN} \tag{11}$$

$$MCC = \frac{TP \times TN + FP \times FN}{\sqrt{(TP + FP)(TN + FN)(TP + FN)(TN + FP)}} \tag{12}$$

$$SN = \frac{TP}{TP + FN} \tag{13}$$

$$SP = \frac{TN}{TN + FP} \tag{14}$$

where, TP (true positives) and TN (true negatives), respectively denote the number of the correctly predicted 6mA and non-6mA samples; FP (false positives) and FN (false negatives) are the number of the falsely predicted non-6mA and 6mA samples, respectively. In addition, we utilized the receiver operating characteristic (ROC) curves to visually test the prediction performance of HD-6mAPred. AUC is computed as the area beneath the ROC curve.

## RESULTS

### Selection of the optimal feature combination

We conducted a series of five-fold cross-validations on the *Rosaceae* training set using both the BiGRU and CNN models to identify the optimal feature combinations. ACC and AUC values of various features and their corresponding combinations were compared.

**Table 1 Performance comparison of different feature encoding ways for BiGRU model.**

| Feature encoding ways | ACC | AUC |
|---|---|---|
| One-hot | 0.520 | 0.521 |
| EIIP | 0.498 | 0.500 |
| NCP | 0.498 | 0.500 |
| ENAC | 0.903 | 0.957 |
| ENAC+one-hot | 0.914 | 0.831 |
| ENAC+EIIP | 0.822 | 0.649 |
| ENAC+NCP | 0.817 | 0.640 |
| ENAC+one-hot+EIIP | 0.733 | 0.669 |
| ENAC+one-hot+NCP | **0.920** | **0.860** |
| ENAC+one-hot+NCP+EIIP | 0.857 | 0.716 |

Note:
Values in bold indicate the best results for each metric.

**Table 2 Performance comparison of different feature encoding ways for CNN model.**

| Feature encoding ways | ACC | AUC |
|---|---|---|
| One-hot | 0.934 | 0.976 |
| EIIP | 0.913 | 0.963 |
| NCP | 0.940 | 0.979 |
| ENAC | 0.930 | 0.976 |
| NCP+one-hot | 0.938 | 0.979 |
| NCP+EIIP | 0.939 | 0.979 |
| NCP+ENAC | **0.943** | **0.981** |
| NCP+ENAC+one-hot | 0.940 | 0.980 |
| NCP+ENAC+EIIP | 0.942 | 0.981 |

Note:
Values in bold indicate the best results for each metric.

As presented in Table 1, the BiGRU model demonstrates that ENAC encoding yields the highest ACC of 0.903 and AUC of 0.957 for a single feature. Consequently, we selected ENAC encoding as our first-order feature. Subsequently, we examined the combination of ENAC encoding with the other three feature encoding ways: one-hot, EIIP and NCP. Among the combinations, ENAC combined with one-hot (denoted as ENAC+one-hot) achieved the highest ACC of 0.914 and AUC of 0.831. Accordingly, ENAC+one-hot is selected as the optimal second-order feature. Further investigations into the combination of ENAC+one-hot with other single features illustrates that ENAC+one-hot+NCP has the highest ACC of 0.920 and AUC of 0.860, but those of them in ENAC+one-hot+NCP+EIIP are descreased. Therefore, the third-order feature ENAC+one-hot+NCP is selected as the optimal feature for the BiGRU model.

For the CNN model, as shown in Table 2, NCP encoding provided the highest ACC (0.940) and AUC (0.979) when considered as a single feature. Therefore, we chose NCP encoding as our first-order feature. We then explored the combination of NCP with the other three encodings: one-hot, EIIP and ENAC. Among the combinations, NCP

combined with ENAC (denoted as NCP+ENAC) achieved the highest ACC and AUC values. Accordingly, NCP+ENAC is selected as the optimal second-order feature. Further investigations into the combination of NCP+ENAC with other single features revealed that when combining all three encoding ways, the highest ACC value achieved is 0.942, which is lower than the 0.943 obtained from NCP+ENAC. Therefore, NCP+ENAC is chosen as the optimal feature combination for the CNN model.

## Motif analysis

To analyze position-specific differences in 6mA and non-6mA-containing sequences for *Rosaceae*, rice and *Arabidopsis*, the two-sample logo software (*Vacic, Iakoucheva & Radivojac, 2006*) at a level of $p < 0.05$ was used. As shown in Fig. 2, the 'A' nucleotide is at the 21st position in 41-nt DNA sequences.

Figure 2 reveals distinct over-representation (enrichment) and under-representation (depletion) of nucleotides at specific positions. For *Rosaceae*, the 'A' base is enriched at positions 12, 17–19, 25 and 28 but depleted at 22, 24, 2 and 30. The 'G' base is enriched at positions 22, 23 and 26, while depleted at 27 and 28. The 'C' base is over-represented at positions 24, 27 and 30 but under-represented at 20, 25, 28 and 29. In rice, the 'A' base is enriched at positions 15–18, 25, and 28. The 'C' base is more abundant at positions 19, 23, 27 and 30. The 'G' base is over-represented at 13, 20, 26 and 29 and under-represented at 27. For *Arabidopsis*, 'A' is enriched at positions 15–18, 20, 25, 29 and 32; 'C' at 19 and 27; 'G' at 20, 22–24, and 28; and 'T' is depleted at 14–20, 25–26, and 37–41. Given these sequences containing 6mA are enriched and depleted with some nucleotides at some positions, it is speculated that the sequence information contribute in discriminating the 6mA from non-6mA.

## Model analysis

To verify the effectiveness of combining BiGRU, CNN, and an attention mechanism in improving predictive performance, we compared six different models using their corresponding optimal encoding ways on the independent testing sets of *Rosaceae*, rice and *Arabidopsis*. Among these models, one model is our proposed HD-6mAPred, the other five models are BiGRU, CNN, the combination of BiGRU and CNN (denoted as BiGRU+CNN), and the two models that incorporate attention mechanisms, referred to as BiGRU+Attention and CNN+Attention, respectively.

Table 3 presents a comprehensive performance comparison of these models. Notably, the HD-6mAPred exhibited superior performance across all testing species. Specifically, for *Rosaceae*, HD-6mAPred achieved an ACC value of 0.996, outperforming BiGRU by 0.481, CNN by 0.050, BiGRU+Attention by 0.414, CNN+Attention by 0.064, and BiGRU+CNN by 0.050. For rice, the ACC value for HD-6mAPred is 0.952, which is 0.014 higher than BiGRU, 0.016 higher than CNN, 0.015 higher than BiGRU+Attention, 0.024 higher than CNN+Attention, and 0.016 higher than BiGRU+CNN. In *Arabidopsis*, HD-6mAPred achieved an ACC of 0.937, exceeding BiGRU by 0.100, CNN by 0.074, BiGRU+Attention by 0.082, CNN+Attention by 0.085, and BiGRU+CNN by 0.074. Furthermore, the

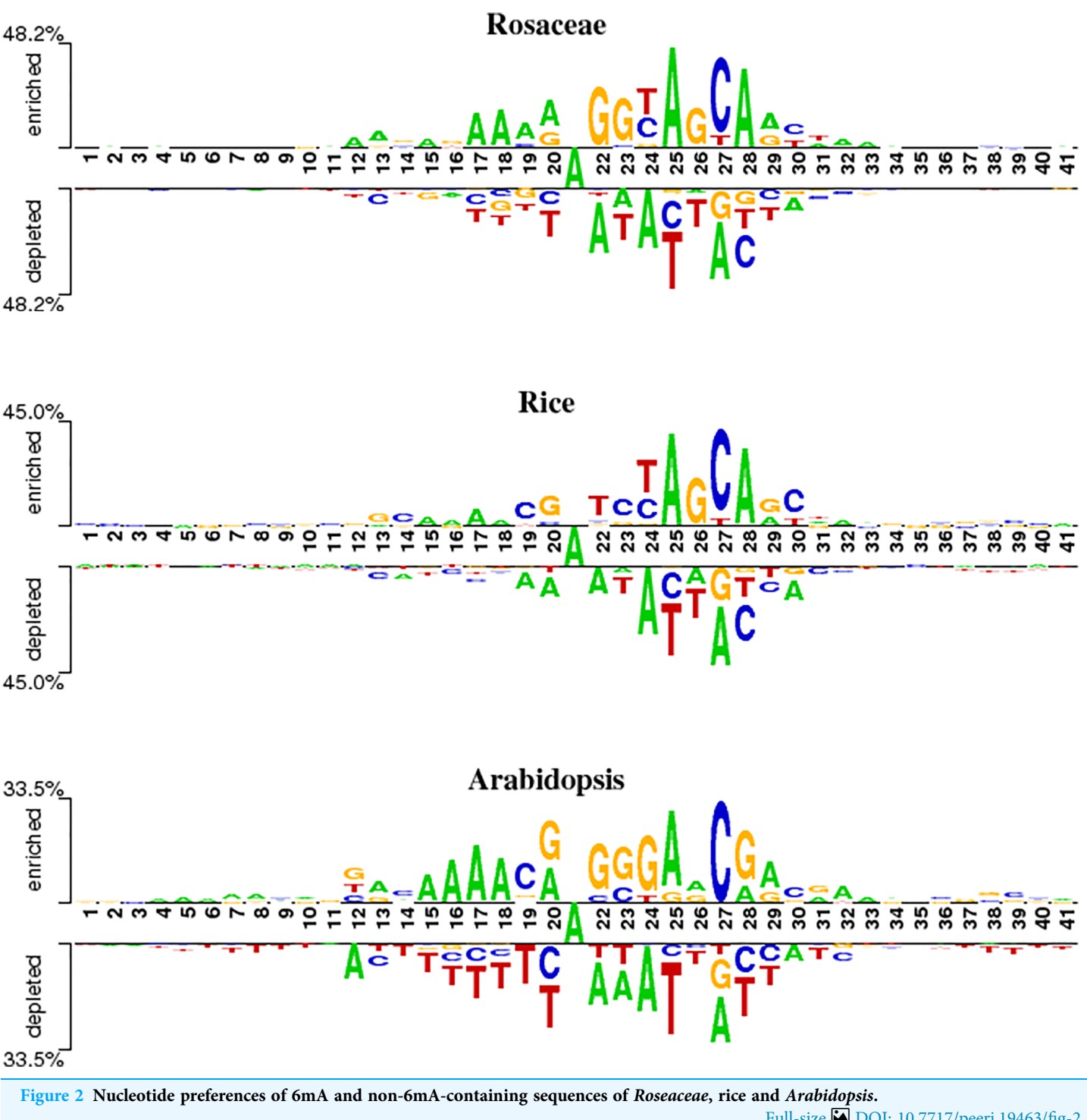

**Figure 2  Nucleotide preferences of 6mA and non-6mA-containing sequences of *Roseaceae*, rice and *Arabidopsis*.**

HD-6mAPred demonstrated significant improvements in MCC, SN and SP, indicating its robust capability to accurately identify both true positive and true negative 6mA sites. The ROC curves, illustrated in Fig. 3, further support these findings, with AUC values for

**Table 3 Comparative results of the six models on the independent test datasets.**

| Species | Feature encoding ways | Models | ACC | MCC | SN | SP |
|---|---|---|---|---|---|---|
| *Rosaceae* | ENAC+one-hot+ NCP | BiGRU | 0.515 | 0.029 | 0.489 | 0.540 |
| | NCP+ENAC | CNN | 0.946 | 0.892 | 0.949 | 0.943 |
| | ENAC+one-hot+ NCP | BiGRU+Attention | 0.582 | 0.161 | 0.381 | 0.781 |
| | NCP+ENAC | CNN+Attention | 0.932 | 0.864 | 0.934 | 0.930 |
| | OEERM | BiGRU+CNN | 0.946 | 0.892 | 0.947 | 0.945 |
| | OEERM | HD-6mAPred | **0.996** | **0.993** | **0.995** | **0.998** |
| Rice | ENAC+one-hot+ NCP | BiGRU | 0.938 | 0.875 | 0.943 | 0.932 |
| | NCP+ENAC | CNN | 0.936 | 0.871 | 0.945 | 0.926 |
| | ENAC+one-hot+ NCP | BiGRU+Attention | 0.937 | 0.874 | 0.944 | 0.931 |
| | NCP+ENAC | CNN+Attention | 0.928 | 0.857 | 0.939 | 0.918 |
| | OEERM | BiGRU+CNN | 0.936 | 0.872 | 0.948 | 0.924 |
| | OEERM | HD-6mAPred | **0.952** | **0.905** | **0.955** | **0.949** |
| *Arabidopsis* | ENAC+one-hot+ NCP | BiGRU | 0.837 | 0.674 | 0.823 | 0.851 |
| | NCP+ENAC | CNN | 0.863 | 0.727 | 0.852 | 0.874 |
| | ENAC+one-hot+ NCP | BiGRU+Attention | 0.855 | 0.709 | 0.836 | 0.873 |
| | NCP+ENAC | CNN+Attention | 0.852 | 0.704 | 0.838 | 0.865 |
| | OEERM | BiGRU+CNN | 0.863 | 0.726 | 0.851 | 0.875 |
| | OEERM | HD-6mAPred | **0.937** | **0.875** | **0.927** | **0.948** |

Note:
"OEERM" represents adopting the corresponding optimal encoding for each module in the model. For example, CNN adopts NCP+ENAC, and BiGRU adopts ENAC+one-hot+ NCP. Values in bold indicate the best results for each metric.

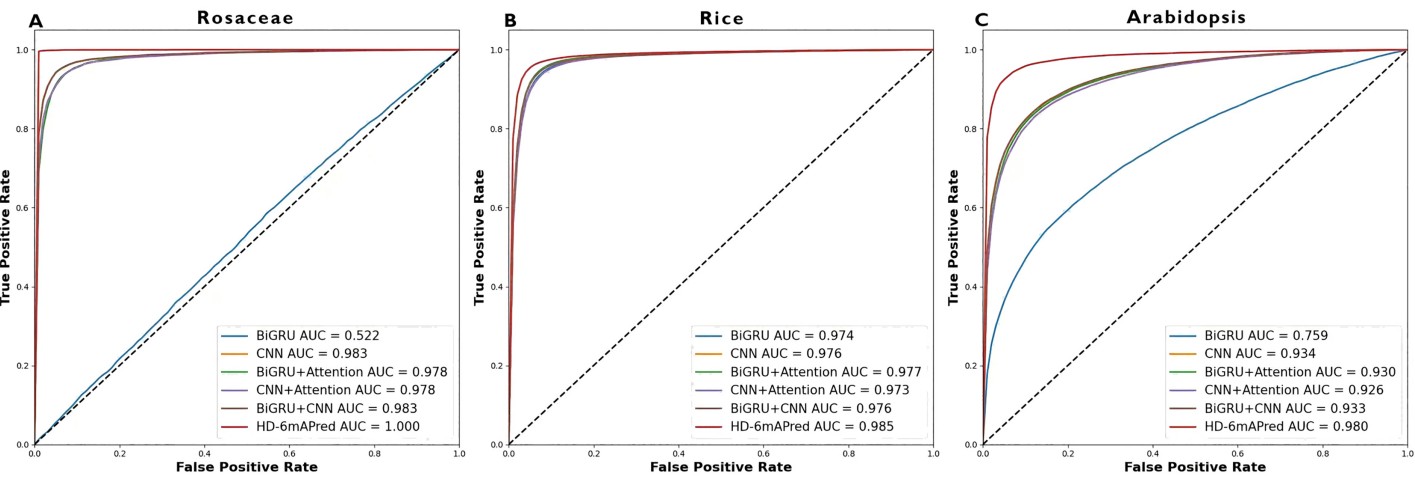

**Figure 3 ROC curves of different models on the three independent test datasets.** (A) ROC curve on the *Rosaceae* dataset; (B) ROC curve on the Rice dataset; (C) ROC curve on the *Arabidopsis* dataset.

HD-6mAPred being 1.000 for *Rosaceae*, 0.985 for rice, and 0.980 for *Arabidopsis*. These results demonstrate the effectiveness of employing a hybrid model that integrates BiGRU, CNN and the attention mechanism to enhance predictive performance for 6mA site identification.

## Performance comparison with existing methods

In recent years, several excellent computational methods have been developed to identify 6mA sites. We conducted a comparison of our HD-6mAPred approach with these existing methods using 5-fold cross-validation across three independent testing sets: the *Rosaceae* testing set, the Rice testing set and the *Arabidopsis* testing set. Table 4 lists the comparison methods and results. We complied the code of CNN6mA (*Tsukiyama, Hasan & Kurata, 2022*) on our test datasets, and the values of evaluation metrics for other existing methods were directly cited from the results reported by i6mA-Caps (*Rehman et al., 2022*) or 6mA-StackingCV (*Huang, Huang & Luo, 2023*). As shown in Table 4, among all the comparison methods, 6mA-StackingCV and i6mA-Caps are the latest and the best methods, respectively, for predicting 6mA sites in the three plant species. Therefore, we mainly reported the comparison results with these two methods.

On the *Rosaceae* dataset, HD-6mAPred demonstrates the most excellent results, surpassing all other approaches. Its values of ACC, MCC, SN and SP are 0.996, 0.993, 0.995 and 0.998, respectively. Compared with 6mA-StackingCV, they are 0.036, 0.073, 0.036 and 0.037 higher, respectively; compared with i6mA-Caps, they are 0.029, 0.055, 0.029 and 0.03 higher, respectively.

On the Rice dataset, although the SN value (0.955) of HD-6mAPred is slightly lower than that of several methods, including Meta-i6mA, iDNA6mA-Rice, MM-6mAPred, 6mA-vote, 6mA-StackingCV and CNN6mA, and its MCC value (0.905) is lower than that of CNN6mA (0.995), HD-6mAPred performed best in the other two evaluation metrics. Its ACC and SP are 0.952 and 0.949, respectively. Compared with 6mA-StackingCV, these two metrics are 0.107 and 0.223 higher, respectively. In addition, compared with i6mA-Caps, HD-6mAPred increased the ACC by 0.012, the MCC by 0.025, the SN by 0.004, and the SP by 0.02.

On the *Arabidopsis* dataset, HD-6mAPred performed best in three of the four evaluation metrics, and its SP value (0.948) is almost the same as that of i6mA-Fuse_FV (0.949). The values of ACC, MCC, SN and SP of HD-6mAPred are 0.937, 0.875, 0.927 and 0.948, respectively. Compared with 6mA-StackingCV, these values are increased by 0.155, 0.299, 0.250 and 0.082, respectively; compared with i6mA-Caps, they are increased by 0.069, 0.135, 0.105 and 0.033, respectively.

To visually highlight the superiority of HD-6mAPred, we presented a performance comparison between HD-6mAPred and the latest methods 6mA-StackingCV, as well as the currently best methods i6mA-Caps by bar charts. The results are illustrated in Fig. 4. Overall, our method achieved the best performance in almost all evaluation metrics. These results not only demonstrate the excellent predictive capabilities of HD-6mAPred, but also evidence its strong generalization ability across different species.

In addtion, to estimate the performance of the model on imbalanced datasets, we randomly extracted 10% of the positive samples from each species while keeping the number of negative samples unchanged, constructing imbalanced datasets with a ratio of 1:10 for the three species, and then tested the performance of our model on these datasets through 5-fold cross-validation. And we obtained some intresting results. For example, for

| Table 4 Comparison results with existing methods. | | | | | |
|---|---|---|---|---|---|
| Species | methods | ACC | MCC | SN | SP |
| *Rosaceae* | Meta-i6mA* | 0.953 | 0.905 | 0.954 | 0.951 |
| | i6mA-Fuse_FV* | 0.943 | 0.887 | 0.924 | 0.962 |
| | i6mA-Fuse_RC* | 0.893 | 0.786 | 0.890 | 0.895 |
| | i6mA-stack_FV* | 0.928 | 0.856 | 0.928 | 0.927 |
| | i6mA-stack_RC* | 0.899 | 0.798 | 0.920 | 0.877 |
| | i6mA-Pred* | 0.840 | 0.684 | 0.897 | 0.782 |
| | iDNA6mA-Rice* | 0.878 | 0.764 | 0.951 | 0.805 |
| | MM-6mAPred* | 0.873 | 0.758 | 0.961 | 0.785 |
| | 6mA-Finder* | 0.846 | 0.701 | 0.928 | 0.764 |
| | 6mA-vote* | 0.955 | 0.909 | 0.955 | 0.954 |
| | 6mA-StackingCV* | 0.960 | 0.920 | 0.959 | 0.961 |
| | i6mA-Caps* | 0.967 | 0.938 | 0.966 | 0.968 |
| | CNN6mA | 0.959 | 0.920 | 0.955 | 0.965 |
| | HD-6mAPred | **0.996** | **0.993** | **0.995** | **0.998** |
| Rice | Meta-i6mA* | 0.880 | 0.768 | 0.957 | 0.802 |
| | i6mA-Fuse_FV* | 0.890 | 0.781 | 0.921 | 0.859 |
| | i6mA-Fuse_RC* | 0.775 | 0.571 | 0.907 | 0.644 |
| | i6mA-stack_FV* | 0.876 | 0.756 | 0.938 | 0.815 |
| | i6mA-stack_RC* | 0.813 | 0.640 | 0.915 | 0.712 |
| | i6mA-Pred* | 0.791 | 0.592 | 0.878 | 0.705 |
| | iDNA6mA-Rice* | 0.755 | 0.561 | 0.960 | 0.547 |
| | MM-6mAPred* | 0.834 | 0.689 | 0.958 | 0.710 |
| | 6mA-Finder* | 0.809 | 0.636 | 0.928 | 0.690 |
| | 6mA-vote* | 0.882 | 0.774 | 0.961 | 0.803 |
| | 6mA-StackingCV* | 0.845 | 0.710 | **0.963** | 0.726 |
| | i6mA-Caps* | 0.940 | 0.880 | 0.951 | 0.929 |
| | CNN6mA | 0.812 | **0.995** | 0.962 | 0.661 |
| | HD-6mAPred | **0.952** | 0.905 | 0.955 | **0.949** |
| *Arabidopsis* | Meta-i6mA* | 0.787 | 0.600 | 0.636 | 0.936 |
| | i6mA-Fuse_FV* | 0.749 | 0.542 | 0.545 | **0.949** |
| | i6mA-Fuse_RC* | 0.757 | 0.534 | 0.615 | 0.897 |
| | i6mA-stack_FV* | 0.770 | 0.570 | 0.604 | 0.933 |
| | i6mA-stack_RC* | 0.751 | 0.514 | 0.634 | 0.865 |
| | i6mA-Pred* | 0.730 | 0.462 | 0.679 | 0.780 |
| | iDNA6mA-Rice* | 0.734 | 0.473 | 0.655 | 0.812 |
| | MM-6mAPred* | 0.765 | 0.531 | 0.784 | 0.747 |
| | 6mA-Finder* | 0.724 | 0.448 | 0.741 | 0.706 |
| | 6mA-vote* | 0.798 | 0.617 | 0.666 | 0.929 |
| | 6mA-StackingCV* | 0.782 | 0.576 | 0.677 | 0.866 |
| | i6mA-Caps* | 0.868 | 0.740 | 0.822 | 0.915 |
| | CNN6mA | 0.763 | 0.532 | 0.686 | 0.839 |
| | HD-6mAPred | **0.937** | **0.875** | **0.927** | 0.948 |

Note:
The asterisk (*) indicates that the results were from the i6mA-Caps (*Rehman et al., 2022*) or 6mA-StackingCV (*Huang, Huang & Luo, 2023*). Values in bold and underlined indicate the best and the second best results for each metric, respectively.

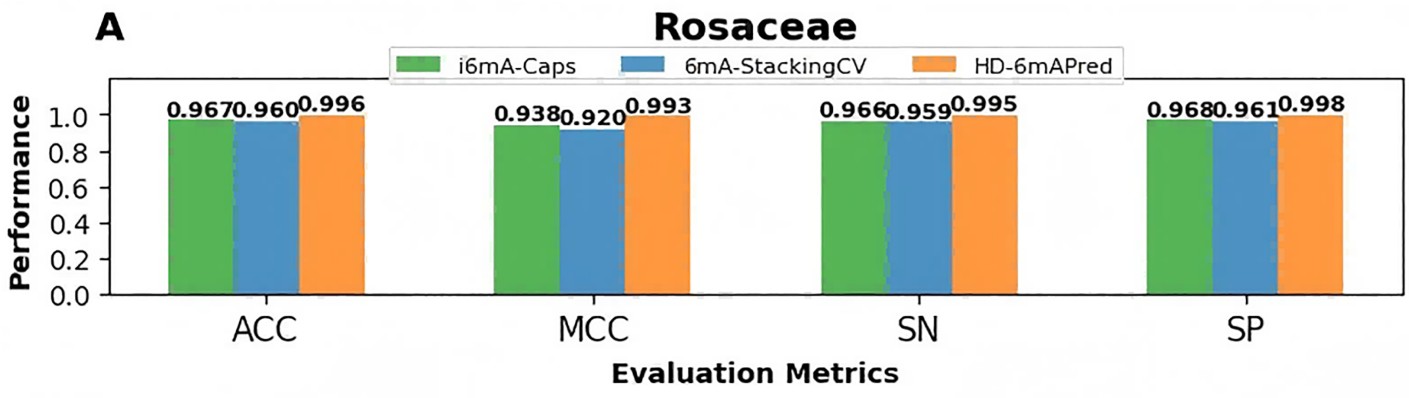

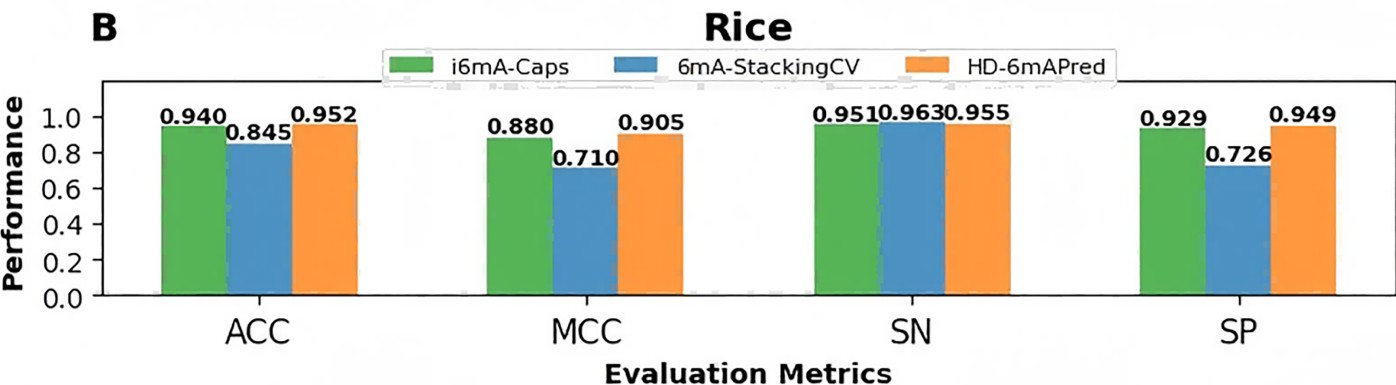

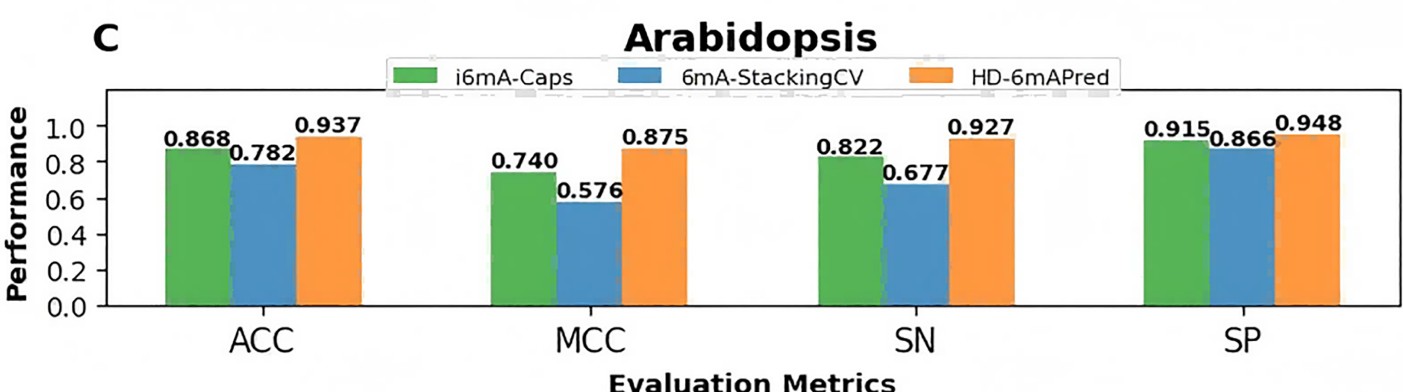

**Figure 4 Performance comparison between HD-6mAPred and the latest method 6mA-StackingCV, as well as the currently best method i6mA-Caps in the three independent test datasets.** (A) Performance of different methods on the *Rosaceae* dataset; (B) performance of different methods on the Rice dataset; (C) performance of different methods on the *Arabidopsis* dataset.

*Rosaceae*, ACC, MCC, SN, SP and AUC were 0.983, 0.966, 0.984, 0.982, and 0.997, respectively; for rice, the corresponding values were 0.954, 0.908, 0.958, 0.949, and 0.986, respectively. For *Arabidopsis*, the corresponding values were 0.928, 0.856, 0.915, 0.940, and

0.672, respectively. These results further demonstrate the robust generalization ability of the HD-6mAPred model.

## DISCUSSION

Feature extraction and the choice of learning algorithms are two key factors in successfully predicting 6mA sites. In this study, we presented a hybrid deep learning approach, named HD-6mAPred, for predicting 6mA sites. The method integrates four different DNA sequence encoding schemes, including one-hot encoding, EIIP, NCP and ENAC,as well as three advanced deep learning models: BiGRU, CNN, and attention mechanism. We employed a hold-out search strategy to identify the optimal sequence encoding ways under the CNN and BiGRU frameworks. Following this, DNA sequences were converted into feature matrices based on their corresponding encoding strategies and input into the respective deep learning algorithms. The outputs from the BiGRU and CNN models were assigned importance weights by an attention mechanism before being processed through a fully connected layer. The proposed HD-6mAPred method not only considers the chemical properties and nucleotide composition of DNA sequences during encoding, but also accounts for long-distance dependencies within sequences, local sequence features, and the relative importance of different feature information, effectively improving the prediction performance for DNA 6mA sites identification.

As a result, HD-6mAPred reached the most excellent performance in all three plant species including *Rosaceae*, rice and *Arabidopsis*, and outperformed the existing methods. In the *Rosaceae* dataset, HD-6mAPred outperformed the latest 6mA prediction method, 6mA-StackingCV (*Huang, Huang & Luo, 2023*), with improvements of 3.6% in ACC, 7.3% in MCC, 3.6% in SN, and 3.7% in SP. On the Rice dataset, HD-6mAPred exceeded 6mA-StackingCV (*Huang, Huang & Luo, 2023*) by 10.7% in ACC, 19.5% in MCC, and 22.3% in SP; the SN value is slightly less than 0.8%. For *Arabidopsis*, HD-6mAPred outperformed 6mA-StackingCV (*Huang, Huang & Luo, 2023*), yielding substantial improvements of 15.5% in ACC, 29.9% in MCC, 25% in SN, and 8.2% in SP. The feature selection approach of HD-6mAPred is inspired by 6mA-stacckingCV (*Huang, Huang & Luo, 2023*), however, the key difference lies in HD-6mAPred optimizing the combinations of encoding methods through deep learning algorithms, whereas 6mA-StackingCV (*Huang, Huang & Luo, 2023*) relies on traditional ML methods. In existing methods, although 6mA-StackingCV (*Huang, Huang & Luo, 2023*) reached the best performance in the *Rosaceae*, it is outperformed by other methods in certain metrics across species, such as 6mA-vote (*Teng et al., 2022*) and i6mA-Fuse-FV (*Hasan et al., 2020*) in rice, as well as 6mA-vote and Meta-i6mA (*Hasan et al., 2021*) in the *Arabidopsis*. In fact, except for our HD-6mAPred, all existing methods could not reach the optimal results in all three species, suggesting that the HD-6mAPred is the most robust compared to other methods. Notably, the performance of HD-6mAPred on the *Arabidopsis* is slightly lower relative to the *Rosaceae* and rice, potentially due to the inherent sequence structure of *Arabidopsis*. Future studies will focus on further enhancing HD-6mAPred to enable more accurate site prediction across a broader range of species.

## CONCLUSIONS

We presented a hybrid model, HD-6mAPred, for predicting 6mA sites in DNA, demonstrating superior performance compared to existing methods for *Rosaceae*, rice and *Arabidopsis*. The construction of HD-6mAPred involved the application of a hold-out feature search strategy, combined with BiGRU, CNN, and an attention mechanism. Theoretically, the hold-out feature search strategy ensures optimal feature encoding is selected, while BiGRU effectively processes sequential characteristics and CNN captures significant local features. Additionally, the attention mechanism allows for the extraction of relevant information based on its importance. Theoretical foundations and comparative analyses validated the excellent performance of HD-6mAPred, illustrating its superiority across the three plant species. In the future, we aim to extend our model to a wider variety of species, further enhancing its performance in 6mA site prediction.

### Funding

This work was supported by the National Natural Science Foundation of China (No. 62462069) and the Graduate Quality Curriculum Construction Project of Yunnan Province (No. Yun Degree [2022] 8). The funders had no role in study design, data collection and analysis, decision to publish, or preparation of the manuscript.

### Grant Disclosures

The following grant information was disclosed by the authors:
National Natural Science Foundation of China: 62462069.
Graduate Quality Curriculum Construction Project of Yunnan Province: No. Yun Degree [2022] 8.

### Competing Interests

The authors declare that they have no competing interests.

### Author Contributions

- Huimin Li conceived and designed the experiments, analyzed the data, authored or reviewed drafts of the article, and approved the final draft.
- Wei Gao conceived and designed the experiments, performed the experiments, prepared figures and/or tables, and approved the final draft.
- Yi Tang conceived and designed the experiments, performed the experiments, authored or reviewed drafts of the article, and approved the final draft.
- Xiaotian Guo conceived and designed the experiments, analyzed the data, prepared figures and/or tables, and approved the final draft.

### Data Availability

The data is available at GitHub: https://github.com/Xiaohong-source/6mA-stackingCV.

The source code is available at zenodo and GitHub:

- https://doi.org/10.5281/zenodo.15355131
- https://github.com/gaowei-source/HD-6mAPred.

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
