# Peer review of "HD-6mAPred: a hybrid deep learning approach for accurate prediction of N6-methyladenine sites in plant species"

_PeerJ, doi:10.7717/peerj.19463_

## Round 0.1 · original submission · Major Revisions

While both reviewers find the paper to be interesting and valuable, they have a number of important concerns, some of them rather deep (e.g. that of performance of the method on real-life non-balanced testing sets or benchmarking against other DL models). Both reviewers note the absence of any code under the provided link.

·

Basic reporting

The paper demonstrates a new method for predicting the presence of 6mA in a sequence, which is superior in performance to previous ones. In this regard, the paper mentions many works by predecessors who solved the same or similar problems. However, there are several questions and suggestions regarding some of the references:
1) The reference in line 79 should introduce the DNA 6mA, but it leads to a recent work that is weakly related to the subject. While 6mA has been known for a very long time, it would be possible to refer to the first works that demonstrated its presence in the DNA of living organisms (doi: 10.1038/2181066a0) or, in particular, eukaryotes (doi: 10.1083/jcb.56.3.697).
2) In lines 123-128, the listed works are described as if they also solved the problem of predicting 6mA, although many of them were designed for completely other tasks. Perhaps it is worth reformulating this sentence.
3) The article by Shen et al. is cited as a reference for the one-hot encoding method, while this method of DNA encoding was not proposed by the authors of the cited work, but was widely used before. As more classic works with this approach, I can suggest the following: (1) doi: 10.1038/nbt.3300, (2) doi: 10.1101/gr.200535.115, (3) doi: 10.1038/nmeth.3547.

Experimental design

1) Among the models provided in the benchmark, it would be correct to see other deep learning models. For example, the i6mA-Caps model (doi: 10.1093/bioinformatics/btac434), which was trained on the same data. You may also consider adding the CNN6mA (doi: 10.1016/j.csbj.2022.12.043) and i6mA-CNN (doi: 10.1109/ISCAS58744.2024.10558061) models to the comparison, although they were used for other organisms, but are applicable to solving the same problem.
2) Descriptions of nucleotide encoding using the one-hot (lines 177-179) and binary (lines 189-190) encoding methods are identical. Could you more clearly indicate the differences between these methods in the text?

Validity of the findings

Provided tables show the developed instrument for 6mA predicting performs better than most of existing ones. However, unfortunately, the provided code repository is empty, so it is not possible to validate the presented results.

Additional comments

Also there are some minor typos – in line 179 there is an extra comma at the end of the sentence. In the description of Figure 1, a space is missing between the words “integrating” and “BiGRU”.

·

Basic reporting

1. The equations (1–5) are not critical as they do not represent original contributions. Therefore, it is recommended to remove them and cite the appropriate references instead. (This pertains to the clarity and necessity of including equations, which aligns with ensuring professional article structure and proper use of references.)


2. The authors did not analyse the motifs or features learned by the proposed model. (This concerns whether the results are self-contained and provide sufficient context to support the hypotheses.)

Experimental design

1. The test datasets are balanced; however, this does not reflect real-world scenarios where negative samples are typically more abundant than positive ones.(This critique addresses the realism and rigor of the experimental design and whether it aligns with high technical standards.)

2. Not all hyperparameters are provided, making it impossible to reproduce the results. (Reproducibility is a fundamental part of describing methods in sufficient detail.)

Validity of the findings

The source code is not accessible through the provided link. (The availability of underlying data (including source code) is essential for validating the results.)

Additional comments

N6-methyladenine (6mA) is a key DNA methylation modification with crucial biological functions, requiring accurate prediction methods. This study introduces HD-6mAPred, a hybrid deep learning model combining BiGRU, CNN, and attention mechanisms, leveraging diverse DNA sequence encoding strategies to enhance prediction accuracy and generalization.

Minor revision:
Removing unnecessary equations streamlines the presentation and focuses on original contributions.
Analyzing learned motifs or features would make the results more self-contained and better contextualized.

Suggested Revisions:

Remove non-original equations and cite the relevant sources.
Provide insights into the motifs or features learned by the model to support the biological relevance of the predictions.

Major revision:
The test datasets are balanced; however, this does not reflect real-world scenarios where negative samples are typically more abundant than positive ones.

Not all hyperparameters are provided, making it impossible to reproduce the results.

Balanced test datasets do not align with realistic data distributions in biological research, potentially biasing results.

Missing hyperparameter details hinder reproducibility, a critical aspect of experimental integrity.

The source code is not accessible through the provided link.

Suggested Revisions:

Reassess the test datasets to include imbalanced scenarios reflecting real-world conditions.

Provide a complete list of hyperparameters used in the model, including their values and rationale.

Ensure the source code is accessible via the provided link, or provide an alternative means to access it.

---

## Round 0.2 · accepted · Accept

The reviewers are satisfied with the revisions, although one of them noticed issues with the organisation of the uploaded code. While this does not preclude me from accepting the paper, I hope the authors will consider improving the code.

·

Basic reporting

I am satisfied with the revisins made.

Experimental design

I am satisfied with the revisions made.

Validity of the findings

The code uploaded after revision is correct and sufficient to verify the results, although it's not perfectly organized and has multiple identical blocks.

·

Basic reporting

Pass

Experimental design

No comment

Validity of the findings

No comment

Additional comments

I truly appreciate the authors for the corrections and improvements, according to the previous comments ,authors improved the manuscript and it's passed from my side.